# Temporal dynamics of the vaginal microbiome and host immune markers before, during, and after metronidazole treatment for bacterial vaginosis

Amanda Williams,[1,2] Jacques Ravel,[1,2] Eric Armstrong,[3] Sanja Huibner,[3] Lindsay Rutt,[1,2] Rupert Kaul,[3,4] Johanna B. Holm[1,2]

**ABSTRACT**   This study analyzed metagenomic and immune marker profiles of seven individuals before, during, and after a 7-day course of metronidazole treatment for bacterial vaginosis (BV). Treatment reduced BV-associated bacteria and immune marker levels, with distinct early (days 1–4) and late (days 5–7) phases. Post-treatment variability in microbial and immune marker profiles demonstrated a rapid resurgence of certain BV-associated bacteria, highlighting the need for additional strategies like probiotics to maintain a healthy vaginal microbiome. The study found significant host and microbial influences on immune response variance, with IP-10 and sEcad highly correlated with the vaginal microbiome. The findings identify the optimal timing for administering live biotherapeutics to restore D-lactic acid-producing *Lactobacillus* species dominance and underscore the complexity of BV infection and treatment response among different people.

**IMPORTANCE** Bacterial vaginosis (BV), a common condition associated with an increased risk of preterm birth and sexually transmitted infections, among others, is characterized by a dysbiotic vaginal microbiome associated with the predominance of a diverse assortment of anaerobic bacterial species. Metronidazole is the first-line treatment recommended by the CDC for BV when patients report symptoms. Despite treatment, BV recurrence is common. There is limited data regarding the effects of oral metronidazole on the vaginal microbiome starting at the initiation of treatment, as most studies have compared measurements taken before and after treatment completion. This study utilized metagenomic sequencing, pan-bacterial qPCR, and immune marker measurements to analyze the longitudinal dynamics of the vaginal microbiome and host immune response before, during, and after metronidazole treatment.

**KEYWORDS**   vaginal microbiota, vaginal dysbiosis, antibiotics, metagenome, inflammatory markers

Bacterial vaginosis (BV) is a condition with associated significant adverse outcomes, including an increased risk of sexually transmitted infections (STIs) (1–7), pelvic inflammatory disease (8), and premature birth (9, 10), and recurrence is often rapid and frequent following antibiotic treatment (11). BV is characterized by a non-optimal vaginal microbiome, in which optimal *Lactobacillus* species are depleted, and a diverse array of anaerobic bacteria, particularly *Gardnerella* species, predominates (12). Rapid shifts in microbial composition can be transient or persistent depending on the individual's microbiome stability (13, 14). Although BV etiology is multifactorial, BV generally stems from disruptions to an optimal vaginal microbiome that lead to dysbiosis (15).

Both symptomatic and asymptomatic BV induce genital pro-inflammatory cytokines, such as interleukin-1ß (IL-1ß) (16), and cause epithelial disruption with the release

**Peer Reviewer** Zhenxin Fan, Sichuan University, Chengdu, China

Address correspondence to Johanna B. Holm, jholm@som.umaryland.edu.

J.R. is co-founder of LUCA Biologics, a biotechnology company focusing on translating microbiome research into live biotherapeutics drugs for women's health.

See the funding table on p. 13.

of soluble E-cadherin (sEcad) (17). The epithelial damage may be directly caused by local inflammation (18), and the resulting increase in cervical epithelial permeability enhances susceptibility to pathogenic microbes responsible for some STIs, such as HIV (19). Furthermore, the loss of *Lactobacillus* leads to a less acidic cervicovaginal pH, creating an environment that favors the growth of these pathogens (20). Comparatively, *Lactobacillus* spp. capable of producing D-lactic acid, such as *L. crispatus*, *L. jensenii*, or *L. gasseri*, exist in a mutualistic relationship with the host. An optimal vaginal microbiome is associated with increased STI protection and immune quiescence, achieved through low-level immune system stimulation, the production of lactic acid (21), and antimicrobial byproducts (15, 22). *L. iners* presents a more complex relationship with reproductive health because, unlike other *Lactobacillus* spp., *L. iners* can only produce L-lactic acid (23). Studies suggest that *L. iners* can persist in BV-associated microbial communities, potentially facilitating shifts between dysbiotic and lactobacilli-dominant states (24–26), and it is frequently associated with an increased risk of BV recurrence, STI acquisition, and negative birth outcomes (27, 28).

The recommended clinical treatment for BV is topical or oral metronidazole (MET) (29). While MET can partially restore a balanced vaginal microbiota, studies have shown that the composition does not fully return to its pre-BV state (30). Following MET treatment, the vaginal microbiota is typically dominated by *L. iners* (11, 31–33). Some people may experience a rebound in beneficial *Lactobacillus* species, leading to the re-establishment of an optimal vaginal microbiome dominated by *L. crispatus*, *L. jensenii*, or *L. gasseri*. However, Mayer et al. (11) observed that although MET treatment led to a notable decrease in BV-associated bacteria and an increase in beneficial *Lactobacillus* species, these changes were often temporary, with many participants experiencing a recurrence of BV, along with a resurgence of the BV-associated bacteria within weeks of completing the treatment. Similarly, Turner et al. found that while *Lactobacillus* species increased in most participants within 4–14 days post-treatment, persistently high levels of *Gardnerella* species Gsp07 were associated with refractory responses to treatment (34). A study examining the effects of a 7-day course of oral MET on the vaginal microbiota of Rwandan women with BV found that treatment resulted in a modest reduction of BV-associated anaerobes and an increase in *L. iners* (35). Yet, the overall cure rate was only 54.5%, with treatment failure associated with higher pre-treatment concentrations of non-optimal species, particularly *G. vaginalis*. These findings were confirmed by Gustin et al. (36), who observed that women with greater vaginal microbiota richness at the time of BV diagnosis were more likely to experience recurrent BV following oral MET treatment. In the case of *L. iners* predominated microbiota, Lee et al. (37) reported that MET efficacy decreases in individuals with higher abundances of *L. iners* relative to *G. vaginalis* before treatment. Ultimately, the diversity of vaginal microbiota prior to treatment and the incomplete restoration of *Lactobacillus* spp. post-treatment leaves the vaginal microbiome vulnerable to BV persistence or recurrence (38, 39).

These studies, among others, underscore the complexity of the vaginal microbiota and the challenges associated with re-establishing and maintaining a *Lactobacillus*-dominated microbiota following antibiotic treatment, regardless of pre-treatment vaginal microbiome composition. A study of South African women treated for BV with a single dose of MET found that while treatment induced short-term shifts in vaginal microbiota and mucosal cytokines, treatment failures led to persistently elevated concentrations of pro-inflammatory cytokines in the genital tract. These persistent inflammatory profiles were associated with an increased risk of HIV acquisition (40). Armstrong et al. (41) observed that treatment of BV with topical MET reduced vaginal inflammation, and these benefits were specifically associated with a loss of BV-associated bacteria. The study, which collected genital immune marker and vaginal metagenomic data before and after MET treatment, found that levels of pro-inflammatory cytokines, chemokines, and soluble immune markers of epithelial barrier disruption were reduced following the 7-day course of antibiotics compared to pre-treatment vaginal concentrations. This was concordant with a reduction in total loads of BV-associated bacteria rather

than an increase in *Lactobacillus* abundance. However, the study did not characterize host and microbiota responses during MET treatment, leaving a critical knowledge gap regarding the immunological and microbiological effects initiated by MET treatment (16).

To address this gap, we leveraged samples from a prior observational study in which participants self-collected daily vaginal swabs over 10 weeks to examine how host immunology, microbiomes, and host immunology-microbiome interactions change throughout BV treatment. In this study, we characterized vaginal immune markers and metagenomes from daily samples collected before, during, and after a 7-day course of oral MET (500 BID) in seven participants who experienced clinical BV while in the study. Both the immune marker and microbiome composition exhibited inter- and intra-personal variation, reflecting the complex interactions between the vaginal microbiome and the human immune system, particularly during BV and MET treatment. Nevertheless, D-lactic acid-producing *Lactobacillus* species increased in abundance during the first 4 days of MET before declining, suggesting a potential window for secondary treatment or adjunctive therapies to enhance *Lactobacillus* restoration and prevent BV recurrence. In addition, host immune marker analysis showed that IP-10 and sEcad exhibited distinct and complementary interaction patterns with *Lactobacillus* and BV-associated species prior to and during symptomatic BV, highlighting their potential as biomarkers for molecular BV diagnosis.

## RESULTS

### Participant characteristics

As previously reported, 25 participants experienced symptomatic BV (SBV) while participating in a 10-week observational study (13). Two vaginal swabs stored dry, collected at time points before, during, and after MET treatment by seven participants (representing nine SBV events) were available for immune marker and metagenomic analyses. For each SBV event, we aimed to analyze swabs collected at specific time points: 2 days before SBV onset, the day of BV diagnosis (BVDX), 3 days during MET treatment, and 2 days post-treatment, equating to 66 data points utilized for this study (Fig. 1). Two participants (UAB003 and UAB128) experienced two SBV events within the original 70-day study period. The mean age of the participants was 24.5 years; five participants self-identified as Black and two as White. One participant, UAB128, experienced vaginal bleeding throughout the study, but the information collected in their daily diary did not indicate the cause of the bleeding. Of the nine SBV events analyzed, eight resulted in a vaginal microbiome dominated by *L. iners* (CST III) for the first 2 days after concluding MET. However, only two participants (UAB130 and UAB135) maintained CST III throughout the end of the study, while the remaining participants fluctuated between CST III and CST IV post-MET. Of note, UAB127 maintained CST IV post-MET.

### Bacterial species abundance changes

Although longitudinal patterns of total bacterial abundance (measured by panbacterial qPCR) varied per participant, the overall bacterial abundances decreased over the 7-day MET treatment course ($R^2 = 0.19$; $\log_2$ fold change = $-1.29$; $P = 0.05$; Table S1; Table S2; Fig. 2). On the day of BV diagnosis (BVDX), the combined estimated absolute abundance of BV-associated taxa (BVT: *Gardnerella* spp., *Sneathia* spp., *Fannyhessea vaginae*, *Ca.* Lachnocurva vaginae, *Prevotella* spp., and *Megasphaera lornae*) was significantly higher than that of D-lactic acid-producing *Lactobacillus* (*L. crispatus*, *L. jensenii*, *L. gasseri*, *L. paragasseri*, and *L. mulieris*; DL), with a $\log_2$FC of 8.01 ($P = 0.0004$; Table S3; Fig. 2). During the first 4 days of MET treatment (early MET), BVT absolute abundance decreased ($\log_2$FC = $-2.16$; $P = 0.038$), while the aggregate estimated absolute abundance of DL increased ($\log_2$FC = 3.07; $P = 0.026$). However, between days 5 and 7 of MET treatment (late MET), DL absolute abundances plateaued ($\log_2$FC = 1.36; $P = 0.19$) and subsequently declined

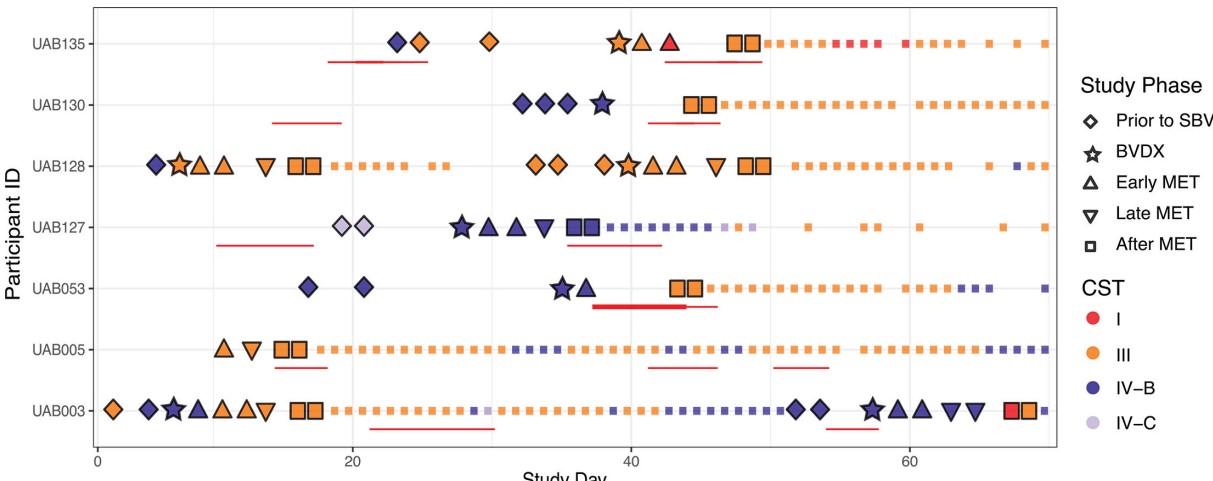

**FIG 1** Study design. Participants clinically diagnosed with BV and prescribed a 7-day oral MET regimen as part of the UMB-HMP study (13) provided daily self-collected vaginal swabs over a 10-week period, with samples selected to represent days preceding symptomatic BV, the day of diagnosis, the MET treatment period, and post-treatment days. The color of each day signifies the CST assigned based on 16S-rRNA sequencing, and shape denotes the study phase. Days for which immunological and metagenomic sequencing data were generated or only 16S-rRNA sequencing was available are indicated by larger and smaller point sizes, respectively. Red lines designate days during which the participant experiences menses.

after treatment completion, returning to levels comparable to pre-treatment but with greater variability between participants. No statistically significant $P$-values were observed after late MET treatment. Notably, DL estimated absolute abundances never surpassed that of the BVT at any point during the study (Table S4). Throughout the study, the estimated absolute abundance of $L.$ $iners$ did not exhibit any significant changes relative to BVT or between study phases. However, by day 6 of MET treatment, $L.$ $iners$ estimated absolute abundance exceeded that of BVT (Table S4). In a regression model, where the independent variable was the study day relative to BVDX, $L.$ $iners$ exhibited the least variation explained by the day of study ($R^2 = 0.20$), followed by BVT ($R^2 = 0.48$), and DL ($R^2 = 0.59$).

Individual species did not necessarily follow the overall trends observed in the combined estimated absolute abundance of BVT or DL (Fig. 3; Fig. S1). Throughout the study, $Lactobacillus$ species exhibited dynamic shifts in estimated absolute abundance across the different phases of BV treatment. Between the pre-SBV phase and BVDX, all $Lactobacillus$ spp. showed either slight decreases or minimal changes, with $L.$ $gasseri$ displaying the most significant decrease ($\log_2$FC = −1.36; $P$ = 0.042). However, from BVDX to the end of early MET treatment, several $Lactobacillus$ species showed increasing trends. $L.$ $crispatus$ significantly increased in abundance with a $\log_2$-fold change of 2.99 ($P$ = 0.029), as did $L.$ $jensenii$ ($\log_2$FC of 1.86, $P$ = 0.03) and $L.$ $gasseri$ ($\log_2$FC = 1.56, $P$ = 0.04). During the late MET phase, trends were mixed, with some species, like $L.$ $gasseri$, exhibiting moderate increases, whereas others, such as $L.$ $mulieris$, declined. By the end of the 7-day MET treatment course, most species either stabilized or experienced a slight decrease in estimated absolute abundance, suggesting that the initial increases in $Lactobacillus$ spp. during early MET treatment were not sustained, and a plateau or reduction occurred post-treatment. Across the entire study, $L.$ $jensenii$ and $L.$ $mulieris$ had the most variation explained by study day during MET treatment ($R^2 = 0.65$ and $R^2 = 0.57$; Table S1).

## Vaginal soluble immune factors

The differential abundance of soluble immune markers was assessed across the study phases (Fig. 4; Table S5). $R^2$ values ranged from 0.18 to 0.51, indicating variability in model fit across different immune markers (Table S6). Among all samples, matrix metallopeptidase 9 (MMP-9) and soluble E-cadherin (sEcad) were the most abundant

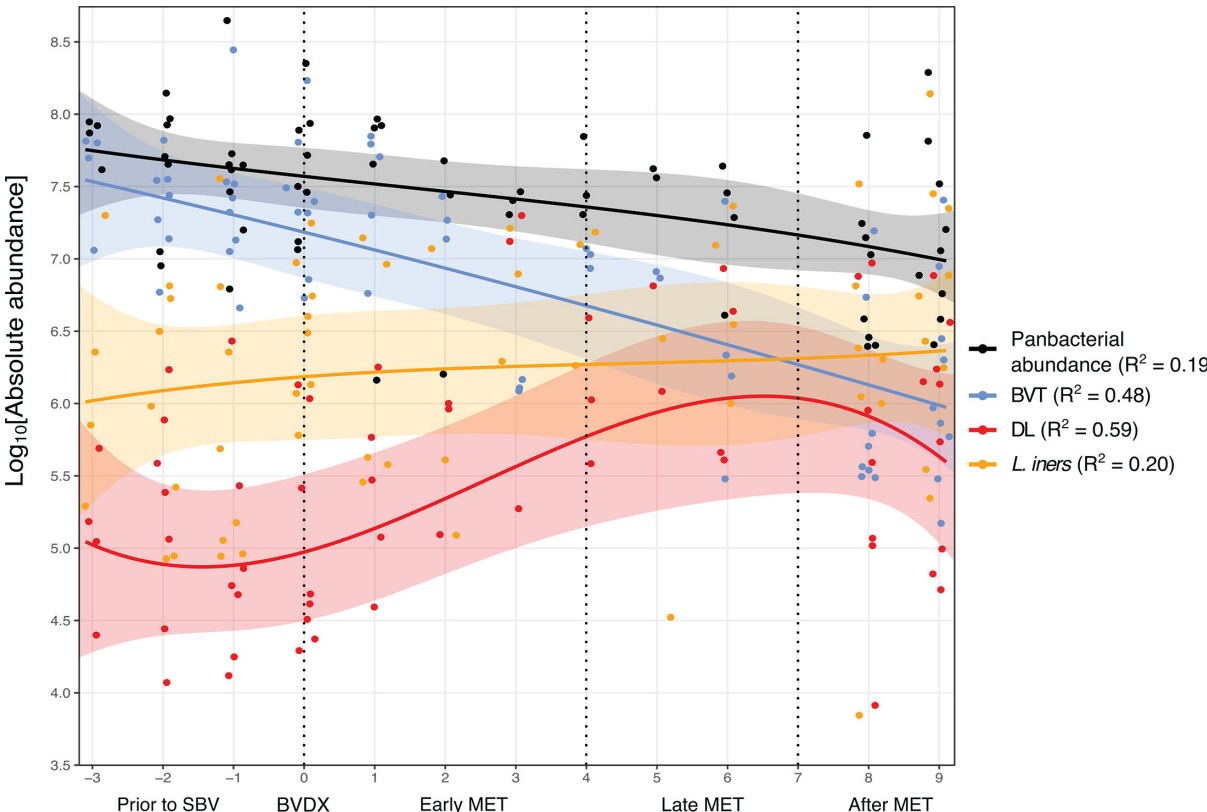

**FIG 2** Bacterial estimated absolute abundance through study phases. The log10 transformed panbacterial total abundances (qPCR; black), in addition to the combined estimated absolute abundance of BVT (*Gardnerella* spp., *Sneathia* spp., *Fannyhessea vaginae*, *Ca*. Lachnocurva vaginae, *Prevotella* spp., and *Megasphaera lornae*; blue), D-lactic acid producing *Lactobacillus* (DL; *Lactobacillus crispatus, L. jensenii, L. gasseri, L. paragasseri*, and *L. mulieris*; red); and *L. iners* (orange) were plotted for all participants through each phase of the pilot study (prior to SBV, BVDX, early MET, late MET, and after MET). Mixed-effects models were generated for each group (solid lines), with 95% confidence intervals applied.

immune markers. From pre-SBV to BVDX, MMP-9 significantly increased (FC = 2.95, $P$ = 0.03), while sEcad concentrations decreased (FC = −0.35, $P$ = 0.01). Throughout and after MET treatment, MMP-9 levels did not significantly change, whereas sEcad concentrations continued to decrease (FC = −1.20, $P$ = 0.04). Pro-inflammatory markers MIP-1β, IL-1α, and IL-1β also showed significant increases from pre-SBV to BVDX, followed by non-significant reductions during MET treatment. After MET treatment completion, MIP-1β and IL-1β exhibited another, though less significant, increase (FC = 0.99, $P$ = 0.06). No immune markers changed significantly between the early and late MET treatment phases.

## Relationships between the vaginal microbiome and immune markers

PERMANOVA tests identified microbiome composition and participant ID as major contributors to the variance observed in immune marker abundances (Table 1). IP-10 and sEcad abundance profiles were primarily driven by the microbiome (46.66% and 43.83%, respectively), while participant ID explained more than 40% of the observed variance of IL-1β, MIG, MIP-3α, IL-6, and IL-1α. Furthermore, sEcad and IL-1α were impacted by the study phase more than any other immune marker (18.46% and 14.53%, respectively).

Given the significant influence of the vaginal microbiome on IP-10 and sEcad, correlations between the concentrations of these immune markers and the estimated absolute abundances of bacterial taxa were examined across the different study phases. Each phase revealed distinct patterns in both the direction and magnitude of these associations. DL species exhibited positive correlations with IP-10 until the completion of MET treatment, whereas BVT species were predominantly negatively correlated during

**TABLE 1** Estimated immune abundance variation[a]

| Immune marker | Observed variation (%) by: | | | |
|---|---|---|---|---|
| | Microbiome | Participant ID | Study phase | Remaining unexplained |
| IP-10 | 46.66 | 19.98 | 10.51 | 22.85 |
| sEcad | 43.83 | 25.05 | 18.56 | 12.56 |
| MIG | 42.17 | 45.70 | 4.61 | 27.52 |
| IL-8 | 37.33 | 36.54 | 2.67 | 23.46 |
| MIP-1β | 36.71 | 35.13 | 1.50 | 26.67 |
| IL-1α | 36.15 | 40.03 | 14.53 | 9.30 |
| IL-17A | 34.73 | 25.50 | 4.82 | 34.96 |
| MMP-9 | 33.51 | 38.67 | 0.99 | 26.83 |
| IFNα2a | 33.40 | 28.67 | 2.23 | 35.70 |
| MIP-3α | 28.81 | 44.33 | 0.85 | 26.01 |
| IL-1β | 18.88 | 52.69 | 3.03 | 25.41 |
| IL-6 | 14.25 | 43.56 | 2.83 | 39.36 |

[a]The observed variation (%) in the immune marker concentrations explained by the vaginal microbiome, participant ID, and phase of BV treatment (i.e., prior to SBV, BVDX, early MET, late MET, and after MET) was calculated by applying PERMANOVA tests to each immune marker. The vaginal microbiome composition was represented by the first 12 principal coordinates to avoid species-species correlations producing overestimations, and significant contributors were summed to get the percent variance explained by the microbiome. The remaining variation not accounted for by the above factors was considered unexplained.

this period (Fig. 5A). Conversely, correlations with sEcad followed an inverse trend: DL species consistently maintained negative associations throughout the study, while BVT species demonstrated positive correlations that persisted until post-MET (Fig. 5B). The strongest correlations were observed during the BVDX and early MET treatment phases. By contrast, the correlations between immune markers and bacterial taxa were attenuated in the late and post-MET treatment phases. Post-MET samples were predominantly characterized by negative correlations between bacterial species and the immune markers analyzed.

## DISCUSSION

The primary treatment for BV has been topical or oral metronidazole since the late 1970s; yet, its efficacy in restoring an optimal vaginal microbiome composition (30) and preventing recurrence remains limited (42). Improved treatments for BV are crucial not only for alleviating symptoms and preventing recurrences but also for reducing its serious consequences, such as increased susceptibility to STIs, urogenital infections, and reproductive complications (1–7, 9, 10, 15, 26). This study employed a high-resolution temporal analysis of the vaginal microbiome and immune marker profiles in seven individuals before, during, and after treatment with oral metronidazole 500 mg twice daily for 7 days, addressing a critical knowledge gap in understanding the effects of BV treatment on the host immune response and the vaginal microbiome composition (43). A key strength of this study was the use of estimated absolute abundance measurements, which provide a more comprehensive view of microbiome dynamics than traditional 16S rRNA gene amplicon sequence-based or metagenomic abundance data. This approach enabled precise comparisons across samples and conditions by accounting for variations in total microbial load.

The effects of MET on the vaginal microbiome composition and soluble immune markers aligned with previous findings; the estimated absolute abundance of BV-associated bacteria decreased, while that of D-lactic acid producing *Lactobacillus* species showed minimal change. In addition, the host immune response was reduced post-MET treatment (41). However, leveraging the study's higher temporal resolution, we identified two distinct phases of MET treatment based on the estimated absolute abundances of key bacterial species: early MET phase (days 1–4) and late MET phase (days 5–7). In particular, DL and *L. iners* estimated absolute abundances increased with a concomitant decrease in most BVT (i.e. *Gardnerella* spp., *Prevoltella* spp., and *Fannyhessa vaginae*).

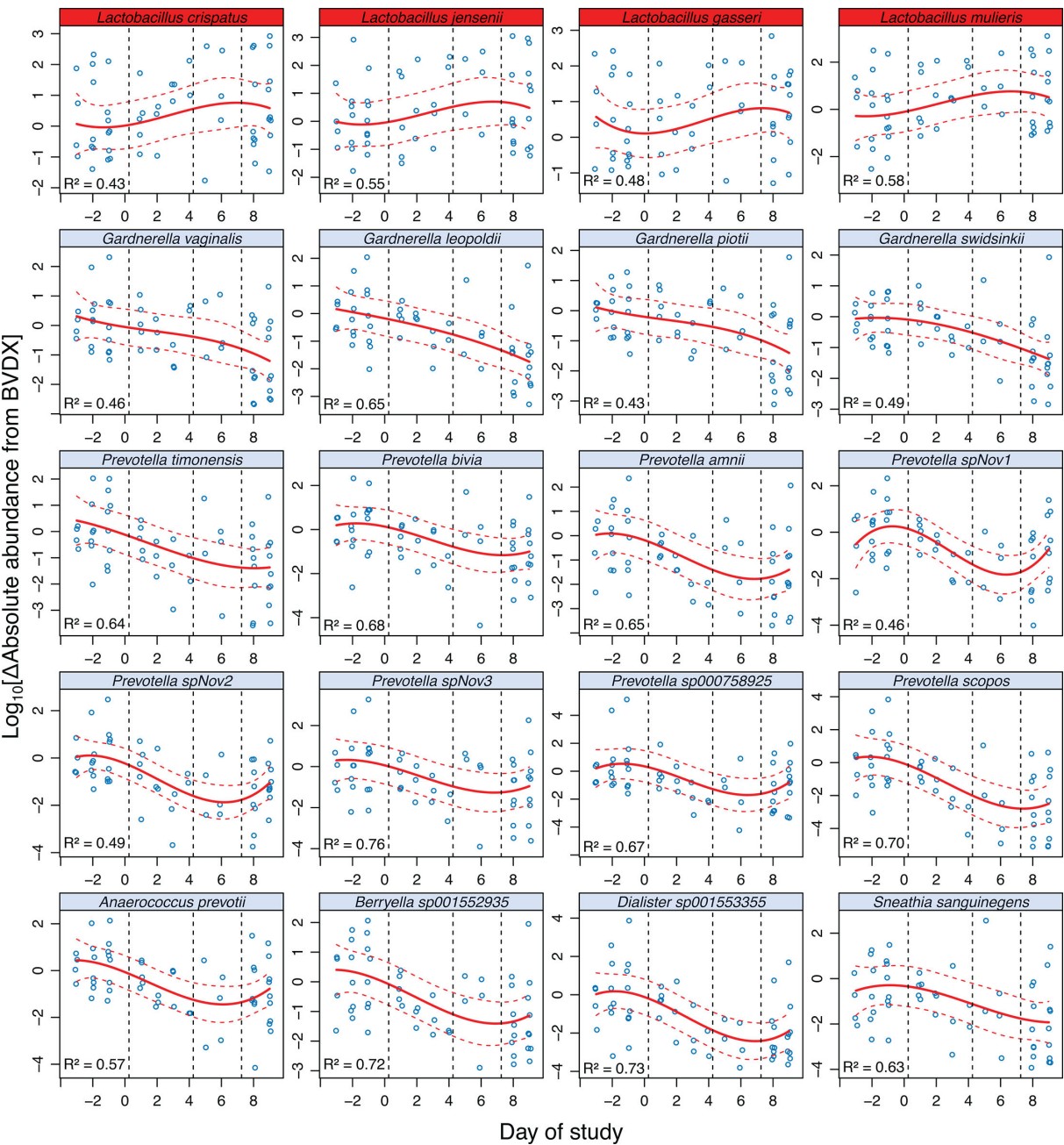

**FIG 3** Bacterial abundances over time. The change in estimated absolute abundance (log10 transformed) from BVDX of key taxa included in DL (red headers) and BVT (light blue headers) for all participants was plotted across the study, with days relative to BVDX, which is represented by day 0. Negative days are those prior to SBV, and day 1 represents the first day the participant started their 7-day course of oral MET. Mixed-effects models (solid red lines) and 95% confidence intervals (dotted red lines) were generated for each species, with the study day as the fixed effect, estimated absolute abundance as the response variable, and participant ID considered a random effect. The conditional $R^2$ values associated with the respective taxa are displayed.

However, other BVT, such as *Megasphaera lorna* and "*Ca*. Lachnocurva vaginae," did not change in abundance during treatment.

The post-treatment period, which has been proposed as a critical period for long-term vaginal health (42), exhibited increased variability in both BVT and *Lactobacillus* species estimated absolute abundances across participants compared to earlier stages of the study, suggesting a reestablishment phase of the microbiome. Notably, several bacterial species exhibited a rapid and pronounced increase shortly after MET treatment ended. These species, *P. amnii*, *Prevotella* sp Nov1, *Prevotella* sp Nov2, *Anaerococcus prevotii*,

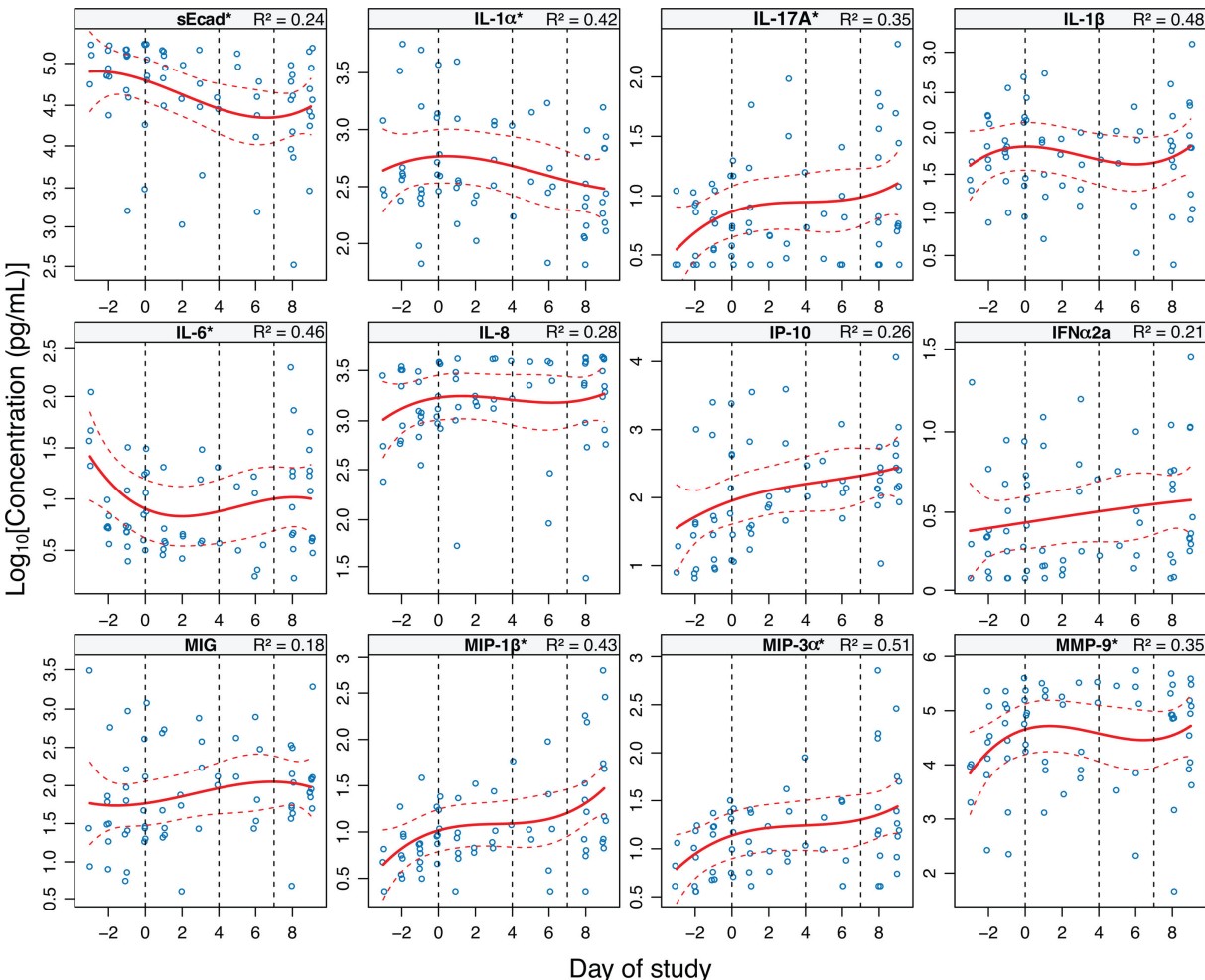

**FIG 4** Immune concentrations over time. The concentrations (pg/mL) of each immune marker for all participants were plotted for each day of the study, with day of study relative to BVDX (day 0). Negative days are those prior to SBV, and day 1 represents the first day the participant started their 7-day course of oral MET. Immune marker concentrations were log10 transformed, and mixed-effects models (solid red lines) and 95% confidence intervals (dotted red lines) were generated for each immune marker, with the study day as the fixed effect, immune marker concentration as the response variable, and participant ID considered a random effect. The conditional $R^2$ values associated with the respective immune marker are present in the plot titles. An asterisk (*) indicates that the immune marker had a $P$-value of <0.05 associated with its differential abundance during any phase of the study.

*Gemella asaccharolytica*, *Berryella* sp001552935, *Dialister micraerophilus*, and *Disalister* sp001553355, represent taxa with diverse and adaptive characteristics. For example, *G. asaccharolytica*, a facultative anaerobe (44), may exhibit variable susceptibility to MET. In general, many of these species show reduced susceptibility to MET (45–47). Meanwhile, *Prevotella* spp., while strict anaerobes, efficiently metabolize host-derived glycans and mucins (48, 49). These substrates may become more accessible as other bacterial populations decline during treatment.

Furthermore, *Gardnerella vaginalis* is a well-established biofilm-forming species, and studies have shown that its biofilms can persist even after treatment with metronidazole (50). Evidence suggests that several taxa can integrate into these polymicrobial biofilms (51–54), where antibiotic penetration is limited, thereby allowing them to persist during treatment and rapidly proliferate post-treatment. Further research is needed to determine whether these taxa contribute to biofilm formation in the vaginal microenvironment. Collectively, these observations suggest that these species possess mechanisms for surviving treatment and rapidly proliferating in the transient ecological niche created by MET treatment-induced shifts in the microbiome.

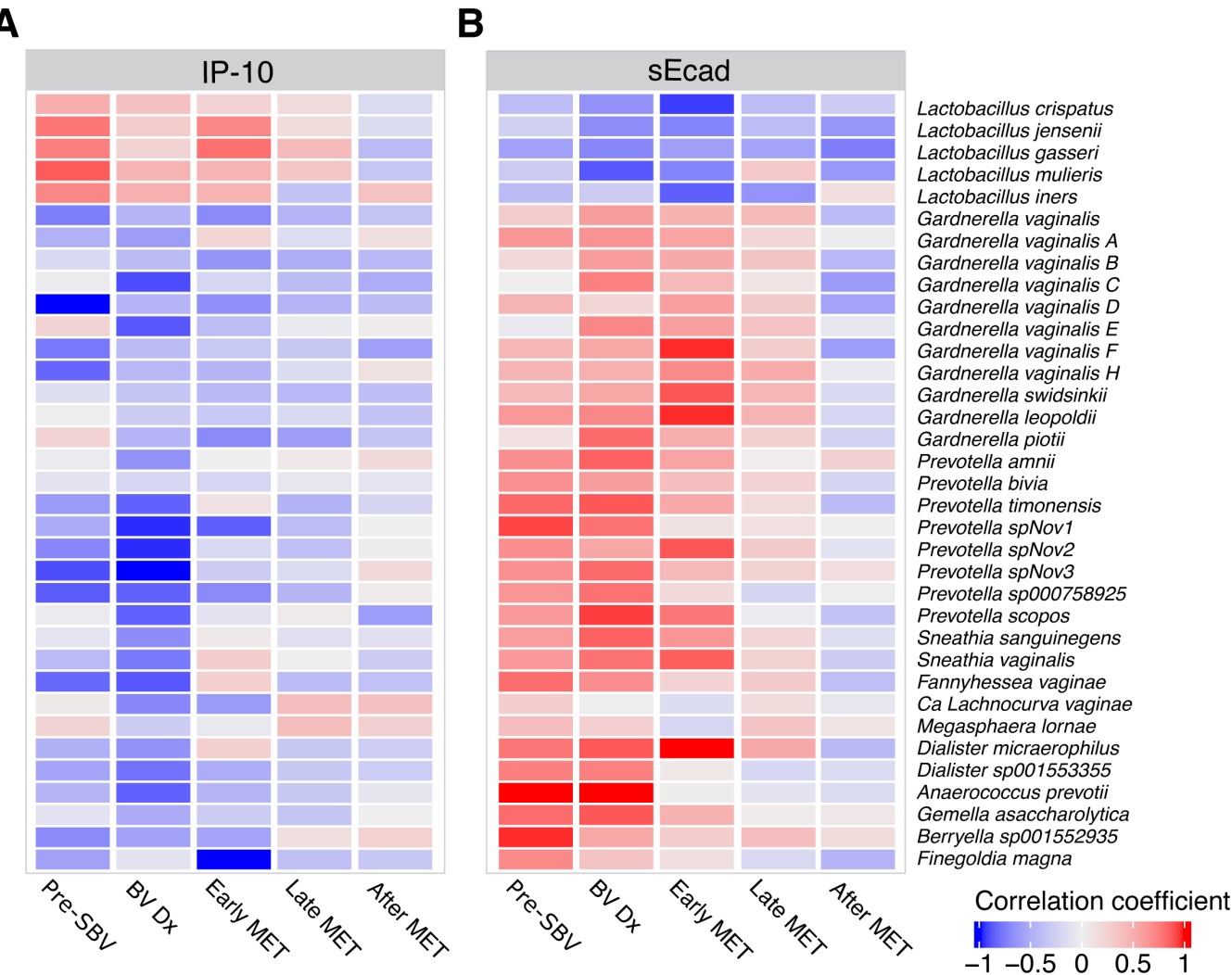

**FIG 5** Correlations between immune markers and the vaginal microbiome taxa. Spearman correlations between concentrations of the immune markers most influenced by the vaginal microbiome (IP-10 and sEcad) and the estimated absolute abundance of key taxa were filtered for significance ($P$-value < 0.05) and plotted across each phase of the study. (A) IP-10 was negatively correlated with many BV-related and lower abundance vaginal species, and (B) sEcad had positive correlations with BV-related species and negative correlations with vaginal *Lactobacillus* species.

Though significant changes in the microbiome were observed during MET treatment, such associations were not quantifiable among most immune markers, underscoring the individualized nature of the vaginal microbiome-host immune interactions (25). When evaluating factors influencing immune response variance, host and microbiome composition were the strongest contributors, with the microbiome exerting a greater influence on IP-10 and sEcad than the host. Interestingly, these immune markers were influenced by the vaginal microbiome more than the host. This underscores the microbiome's role in modulating immune response, consistent with previous studies (40, 55). Although the study phase contributed to immune marker variance in sEcad, IL-1α, and IP-10, its effect was modest compared to host and microbiome factors. However, the substantial unexplained variance in several immune markers, particularly IL-6, IFNα2a, and IL-17A, suggests that additional, unmeasured factors influence immune marker levels.

Recognizing the substantial impact of the vaginal microbiome on IP-10 and sEcad, correlations between these immune markers and the estimated absolute abundances of DL and BVT across study phases were analyzed. The strongest correlations were observed at BVDX and during the early MET treatment phase, signifying that this is

a period of heightened microbiome-immune interactions. By contrast, the attenuation of correlations was attenuated in the late and post-MET treatment phases, possibly reflecting the diminished bacterial load or a transient ecological state dominated by *L. iners*.

More specifically, IP-10 and sEcad demonstrated distinct interaction patterns between DL and BVT species during BV, further supporting their potential as biomarkers for molecular BV diagnosis. Prior to MET treatment completion, DL species were positively correlated with IP-10, supporting previous findings (41, 56) and DL species' role in host anti-inflammatory regulation (7). IP-10 (CXCL10) is a chemokine that plays a key role in immune regulation and defense against pathogens. Its primary function is to recruit immune cells, particularly activated T cells, natural killer cells, and monocytes, to sites of inflammation or infection via binding to its receptor, CXCR3 (57). IP-10 and IL-1β comprise the rapid point-of-care Genital Inflammation Test (GIFT) test for both BV and STIs (58).

By contrast, sEcad exhibited an inverse relationship with DL and BVT. BVT species and sEcad were positively correlated prior to and during MET treatment, while DL species remained negatively associated with sEcad throughout the study. As an essential component of epithelial cell-cell junctions, sEcad is critical for maintaining epithelial barrier integrity that prevents the passage of harmful substances and pathogens (59). The study by Liu et al. (19) identified sEcad as a potential biomarker for vaginal epithelial disruption, indicating that elevated levels of this protein may reflect damage to the genital epithelium and could be used to assess epithelial integrity in reproductive health. However, our findings support using sEcad as a general biomarker for molecular BV diagnosis.

Despite the limited sample size, the study design, which included sampling prior to SBV, at BV diagnosis, during MET treatment, and post-treatment, provided novel insights into the effects of MET on the microbiome composition, immune responses, and their interplay. The microbiome and the individual participant primarily influenced the significant variance in immune marker profiles. While participant variation was evident, sEcad and IP-10 emerged as promising biomarkers of vaginal microbiome dynamics during a symptomatic BV event and MET treatment due to their levels strongly impacted by microbiome factors and their distinct interaction patterns with DL and BVT species.

Understanding the mechanisms driving the rapid resurgence of BV-associated bacteria provides critical insights into post-MET treatment microbial recovery dynamics and informs the development of targeted interventions to prevent recurrence. In this study, no participant transitioned to an optimal and protective *D*-lactic acid-producing *Lactobacillus*-dominated vaginal microbiome (12, 60). Instead, individuals either maintained a CST III microbiome, fluctuated between CST III and CST IV microbiomes, or experienced BV recurrence. These findings underscore the limitations of antibiotic therapy alone and emphasize the need for adjunctive strategies that promote *D*-lactic acid-producing *Lactobacillus* spp. growth during the early phase of MET treatment and foster post-therapy resilience to enhance treatment outcomes (61–66).

Therapies aimed at modulating the composition of the vaginal microbiome, including probiotics combined with MET treatment for BV, have had mixed success but remain promising. A prospective, parallel-group, randomized, controlled study found that oral *L. rhamnosus* GR-1 and *L. reuteri* RC-14 (67) (aka Lactogyn), when administered alongside MET, had no effect on the 30-day BV cure rate compared to MET alone (68). Similarly, EcoVag vaginal probiotic capsules containing *L. gasseri* and *L. rhamnosus*, given after a seven-day course of 2% vaginal clindamycin cream, did not improve the 30-day cure rate (69). The inefficacy of these products may be because the species comprised in these probiotic formulations are not naturally present in the vaginal microbiome (60). By contrast, LACTIN-V, a live biotherapeutic product containing *L. crispatus* CTV-05 (70), significantly reduced BV recurrence when administered vaginally after MET therapy. Participants receiving LACTIN-V were more likely to achieve long-term BV remission than placebo groups (70, 71). Vaginal microbiome transplantation is

another promising approach for introducing beneficial *Lactobacillus* species to promote long-term BV remission in combination with MET therapy (72). Notably, all these studies have attempted to modulate the vaginal microbiome after MET therapy. Interestingly, our study suggests that the optimal window for intervention with probiotics, live biotherapeutics, or VMT may be early in MET treatment when D-lactic acid-producing species naturally increase relative to BV-associated taxa. This finding warrants further investigation.

Comprehensive analyses of the vaginal microbiome and immune factors during MET treatment can reveal important insights into how host-microbe interactions evolve during BV. These insights could help improve both diagnosis and treatment outcomes. This study lays the groundwork for future research to understand the complexities of BV infection and nuanced treatment responses among different people, optimize the live biotherapeutic delivery timing to restore *Lactobacillus* species dominance, and advance biomarker-based BV diagnostics. While the relatively small sample size is a limitation, the study design—featuring longitudinal sampling at key clinical time points—enabled a detailed investigation of dynamic microbial and immune shifts in response to treatment. For improved methods of clearing BV and alleviating its adverse sequelae to become a reality, longitudinal studies with larger cohorts will be essential to validate and expand upon these findings.

## MATERIALS AND METHODS

### Human cohort

This analysis used samples collected between September 2008 and July 2010 from 135 non-pregnant female participants of reproductive age enrolled in a 10 week prospective longitudinal study at the University of Alabama at Birmingham in which two vaginal dry swabs were self-collected daily (13). Participants received clinical evaluations at enrollment, at weeks 5 and 10, as well as when the participant experienced vaginal symptoms. For this secondary analysis, we analyzed swabs surrounding symptomatic BV (SBV) events clinically diagnosed using Amsel's criteria (73) and treated with MET 500 BID for 7 days. Of the original 25 participants who experienced SBV and received treatment, swabs were available for all study phases from seven participants and nine SBV events. For each of these, swabs collected 2 days prior to SBV ($n = 19$), the day of BV diagnosis (BVDX, $n = 8$), 3 days during MET ($n = 20$), and 2 days after MET ($n = 17$) were used to generate genomic DNA extractions, metagenomic sequencing, and immune marker profiles in this pilot study. The swab designated for immune marker profiling was shipped to the University of Toronto. These swabs were collected as part of a previously published study (13).

### Immune marker measurements

Eluates from vaginal swabs in PBS were thawed and centrifuged at 4,500 rpm for 30 minutes prior to soluble immune factor measurement. Supernatant was then removed and analyzed with the MSD platform according to the manufacturer's instructions as previously described (13). Cytokines and soluble immune factors included sEcad, IFNα2a, IL-17A, IL-1α, IL-1β, IL-6, IL-8, IP-10, MIG, MIP-1β, MIP-3α, and MMP-9. Immune marker concentrations were log10 transformed to minimize the effects of highly abundant immune markers.

### DNA extraction, qPCR, and metagenomic sequencing

Genomic DNA was extracted as previously described (13). The total 16S rRNA gene copy number was determined using a PanBacterial qPCR method previously developed by Liu et al. (74). Barcoded genomic DNA libraries were constructed using the NEBNext DNA Sample Prep Master Mix Set 1 (New England Biolabs). Metagenomic libraries were sequenced via Maryland Genomics on a NovaSeq 6000 platform on an S2 Flowcell

and generated 55 million reads per sample. Each sequencing plate included wells with nuclease-free water as a negative control and positive extraction controls using the ZymoBIOMICS Microbial Community Standard. The negative controls were nuclease-free water. Host reads were removed using BMTagger (75) with the GRCh38 human genome as reference. The remaining metagenomic reads were quality filtered using fastp (v.0.21) (76) and then mapped to VIRGO2 (77). The abundance of genes mapped to VIRGO2 (77) was normalized by gene and average library length (gene abundance x 150/gene length). The gene abundance counts for each species were summed to represent the abundance of that species and normalized by sequencing depth. The species counts were then multiplied by the panbacterial qPCR abundance within each sample to get the estimated absolute abundance of each microbial taxon. The data were log10 transformed to minimize the effects of highly abundant species.

## 16S rRNA gene amplicon sequencing data

For this analysis, we also examined existing amplicon sequencing data of the V3V4 regions (PRJNA208535) (13). Taxonomy was assigned using SpeciateIT and vSpeciateDB (78), and CSTs were assigned using VALENCIA (79).

## Statistical analysis

All analytical code can be found at https://github.com/jbholm/BV-Dynamics. Wilcoxon's matched-pairs signed-rank test for each immune marker and microbial taxa was performed using the samples collected on BVDX and the day after the 7-day course of MET was complete with the function wilcox.test (alternative = TRUE, exact = FALSE) as part of the R stats package v4.3.1 (80). The mixed-effects polynomial regression models for the immune marker and microbiome data were produced and tested using the R Statistical Software stats, lme4 v1.1-35.1 (81), performance v.0.12.2 (82), and splines v4.3.1 packages (80). $R^2$ values were obtained using the R package MuMIn v1.47.5 (83). Within the formula, time point was the fixed effect, species or immune marker concentrations were the dependent variables, and participant ID was considered a random effect. Model selection criteria, including Akaike Information Criterion (AIC), Bayesian Information Criterion (BIC), and Log-Likelihood, were calculated using associated functions as part of the stats R package. Residuals were extracted, and the Shapiro-Wilk test was applied to assess the normality of the residuals. Bootstrapping was conducted to assess the robustness of the model's estimates using the bootMer function (81).

Differential abundance analysis was performed for immune markers and taxa between the first day of SBV and BVDX, BVDX and day 4 of MET, day 4 of MET and the final day of MET, and day 7 of MET and the final day of the study. Taxa only present in 10% of samples across both time points were removed. For each retained taxon and immune marker, a two-sample t-test was performed to assess significant differences in abundance between the two treatments. Variable importance in projection (VIP) scores were calculated with the mixOmics v.6.26.0 R package (84).

The immune response variance explained by each factor considered in this study was estimated through permutational multivariate ANOVA (PERMANOVA) tests. The vaginal microbiome composition was represented by the first 12 principal coordinates, which explained >70% of the variation in the microbiome, to avoid species-species correlations producing overestimations. Principal coordinate analysis (PCoA) was performed on the log10-normalized microbiome data using the R package vegan v2.6-4 (85). The input data for the PCoA was a Bray-Curtis dissimilarity matrix, obtained through the vegdist function (method = "bray"). PCoA was performed using the wcmdscale function (eig = TRUE, add = "cailliez"). PERMANOVA tests for each immune marker were conducted using the adonis2 function (permutations = 5,040, method = "bray"). The results were filtered to only include significant contributors ($P < 0.1$), and then the principal coordinates were summed to get the variance caused by the microbial composition, in addition to

participant ID and phase of BV treatment (i.e., prior to SBV, BVDX, early MET, and after MET).

## Correlation analysis

The R stats package v4.3.1 (80) was used to determine whether significant correlations exist between the metagenomic and immune marker data for each stage of BV treatment. The initial species abundance table was filtered before normalization, removing species with a frequency of less than 200 reads across all samples to eliminate low-abundant species that could skew the correlation tests. The normalized estimated absolute abundance data at the species level were used to calculate pairwise correlations with the soluble immune marker profiles using the Spearman method, as well as adjustment of $P$-values using the Benjamini-Hochberg method with the corr.test function. For simplicity, only major *Lactobacillus* spp. and BV-related taxa were included, and correlations were retained if the associated $P$-value was less than 0.05. Correlations were visualized with heatmap plots generated using the Heatmap function in the circlize v0.4.15 R package (86).

## ACKNOWLEDGMENTS

The research reported in this publication was supported by the University of Maryland Baltimore, Institute for Clinical & Translational Research (ICTR) (J.B.H.), and National Institute of Allergy and Infectious Diseases grants from the National Institutes of Health (NIH) under award numbers UH2AI083264 (J.R.) and K01-AI163413 (J.B.H.). A.W. was supported by T32AI162579 from NIAID.

## AUTHOR AFFILIATIONS

[1]Center for Advanced Microbiome Research and Innovation, Institute for Genome Sciences, University of Maryland School of Medicine, Baltimore, Maryland, USA
[2]Department of Microbiology and Immunology, University of Maryland School of Medicine, Baltimore, Maryland, USA
[3]Department of Medicine, University of Toronto, Toronto, Ontario, Canada
[4]Department of Medicine, University Health Network, Toronto, Ontario, Canada

## AUTHOR ORCIDs

Amanda Williams  http://orcid.org/0000-0003-1343-1101
Jacques Ravel  http://orcid.org/0000-0002-0851-2233
Eric Armstrong  http://orcid.org/0000-0002-4530-6631
Rupert Kaul  http://orcid.org/0000-0002-1586-4093
Johanna B. Holm  http://orcid.org/0000-0002-7646-4085

## FUNDING

| Funder | Grant(s) | Author(s) |
| --- | --- | --- |
| University of Maryland Baltimore Institute for Clinical & Translational Research | ID #884 | Johanna B Holm |
| National Institute of Allergy and Infectious Diseases | K01-AI163413 | Johanna B Holm |
| National Institute of Allergy and Infectious Diseases | UH2AI083264 | Johanna B. Holm |
| National Institute of Allergy and Infectious Diseases | T32AI162579 | Johanna B. Holm |

## AUTHOR CONTRIBUTIONS

Amanda Williams, Data curation, Formal analysis, Investigation, Methodology, Software, Validation, Visualization, Writing – original draft, Writing – review and editing | Jacques Ravel, Conceptualization, Formal analysis, Investigation, Methodology, Resources, Supervision, Validation, Writing – original draft, Writing – review and editing |

Eric Armstrong, Conceptualization, Investigation, Methodology, Resources, Validation, Writing – review and editing | Sanja Huibner, Methodology, Resources | Lindsay Rutt, Conceptualization, Formal analysis, Methodology, Resources, Validation, Visualization, Writing – review and editing | Rupert Kaul, Conceptualization, Formal analysis, Investigation, Methodology, Resources, Supervision, Validation, Visualization, Writing – original draft, Writing – review and editing | Johanna B. Holm, Conceptualization, Data curation, Formal analysis, Funding acquisition, Investigation, Methodology, Project administration, Resources, Software, Supervision, Validation, Visualization, Writing – original draft, Writing – review and editing

## DATA AVAILABILITY

The completed STORMS checklist is available at: https://github.com/jbholm/BV-Dynamics/blob/main/STORMS_Excel_1.03.xlsx. All metagenomic data generated in this study have been deposited in the NCBI Sequence Read Archive (SRA) under accession number PRJNA208535 and are publicly accessible.

## ETHICS APPROVAL

The clinical study protocol was approved by the Institutional Review Board of the University of Alabama at Birmingham and the University of Maryland School of Medicine. Written informed consent was appropriately obtained from all participants.

## ADDITIONAL FILES

The following material is available online.

### Supplemental Material

**Fig. S1 (mSystems00380-25-S0001.pdf).** Bacterial abundances over time.
**Table S1 (mSystems00380-25-S0002.csv).** Mixed effects polynomial regression models for the microbiome data.
**Table S2 (mSystems00380-25-S0003.xlsx).** Differential abundance of taxa for each study phase.
**Table S3 (mSystems00380-25-S0004.csv).** Differential abundance of BVT and DL.
**Table S4 (mSystems00380-25-S0005.csv).** Ratio of DL, BVT, and *L. iners* at each time point.
**Table S5 (mSystems00380-25-S0006.xlsx).** Differential abundance of the immune markers for each study phase.
**Table S6 (mSystems00380-25-S0007.csv).** Mixed effects polynomial regression models for the immune marker concentrations.

### Open Peer Review

**PEER REVIEW HISTORY (review-history.pdf).** An accounting of the reviewer comments and feedback.

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
