## [Reviewer comments · mSystems]

Temporal Dynamics of the Vaginal Microbiome and Host Immune Markers Before, During, and After Metronidazole Treatment for Bacterial Vaginosis

Amanda Williams, Jacques Ravel, Eric Armstrong, Sanja Huiber, Lindsay Rutt, Rupert Kaul, and Johanna Holm

Corresponding Author(s): Johanna Holm, University of Maryland School of Medicine

Review Timeline:

Submission Date:	March 19, 2025
Editorial Decision:	April 24, 2025
Revision Received:	May 21, 2025
Accepted:	May 30, 2025

Editor: Ying Zhang

Reviewer(s): Disclosure of reviewer identity is with reference to reviewer comments included in decision letter(s). The following individuals involved in review of your submission have agreed to reveal their identity: Zhenxin FAN (Reviewer #2)

Transaction Report:

DOI: <https://doi.org/10.1128/msystems.00380-25>

Re: mSystems00380-25 (Temporal Dynamics of the Vaginal Microbiome and Host Immune Markers Before, During, and After Metronidazole Treatment for Bacterial Vaginosis)

Dear Dr. Johanna B Holm:

Please address all reviewers' comments and provide a discussion of the small sample size and potential implications. The authors should also consider gaining insights into the microbiome function through a more thorough analysis of the metagenomic data. A clear statement of Data Accessibility is required. Data related to intermediate steps of the data analysis, such as metagenomic assembly and annotation, might be provided to enhance reproducibility of the study.

Revision Guidelines

Sincerely,
Ying Zhang
Editor
mSystems

Reviewer #1 (Comments for the Author):

Williams et al provide new results of paired vaginal microbiome (metagenomes) and cytokines in a human cohort of individuals

presenting with symptomatic bacterial vaginosis that received oral metronidazole treatment. Metronidazole treatment has been the standard of care for symptomatic bacterial vaginosis for over 50 years, however, recurrent BV is common suggesting treatment is insufficient to provide long term resolution. While the exact etiology is not well-understood, BV is thought to arise from a complex dysbiotic interaction between the vaginal microbiota and vaginal immune response that varies across individuals. Although a small cohort ($n = 7$), this study provides novel characterization of the vaginal microbiome, panbacterial abundance, and vaginal cytokine profile longitudinally prior to, during, and after metronidazole treatment. The authors find a total decrease in bacterial abundance over metronidazole treatment, predominantly driven by reduced BV-associated taxa. Concordantly, they observe an increase in D-lactate producing species during treatment, with no significant microbial features at the end of treatment. In line with prior studies, they find an inflammatory vaginal signature upon BV onset. Interestingly, they found that vaginal sEcad and IP-10 levels were primarily explained by microbial features, and their results confirm the utility of sEcad and IP-10 as markers of molecular BV. Overall, the manuscript was well-written and the experimental design and analyses appropriate. Results and discussion incorporate prior work and study limitations. This work contributes meaningfully to the field of bacterial vaginosis and reveals unrecognized host/microbial kinetics during metronidazole treatment. My comments in general are minor and meant to improve the clarity of the work.

Specific comments:

- Metagenomic data was obtained, but relationships beyond taxonomic abundances were not assessed. Were there any functional/genetic associations observed? Can the authors comment on this? How did the genetic signatures of the same CSTs within an individual change over time?
- Figure 1. Consider highlighting the BV Dx a bit more - perhaps an arrow just above the points? While I appreciate the amount of information present in this Figure, which is generally readily interpreted - it took a bit of time to scan for the circle shapes representing BV onset.
- Figure 2. Why were there no day 7 points captured? There appears to be one immune point in Figure 4. Please comment. Also, please indicate in the Figure itself (y-axis) that data is log transformed.
- Figure 4. The figure legend indicates that the data has been log transformed but this is not indicated in the Figure itself - please clarify.
- Figure 5. Were correlations between cytokines and microbial features statistically significant? This could be indicated in Fig. 5 and/or Supp. Table 6.

Reviewer #2 (Comments for the Author):

This study analyzed the dynamics of vaginal microbiota and immune markers in patients with bacterial vaginosis (BV) before, during, and after oral metronidazole (MET) treatment. They revealed that the early treatment is a potential key window for probiotic intervention. In addition, IP-10 and sEcad have been identified as immune markers closely related to the dynamics of BV microbiota, providing new targets for molecular diagnosis and therapeutic monitoring. The results also confirm that metronidazole alone is difficult to restore the protective advantage of lactobacilli, thus emphasizing the necessity of combination therapy (such as probiotics and live biologics).

Major comments:

1. The final sample size used is relatively small, authors should mention this point in the discussion.
2. It is mentioned in the article that the rebound of BV related bacteria after treatment may be related to biofilm, but there is a lack of direct evidence (such as biofilm imaging or gene expression data). It is recommended to supplement relevant discussions or cite supporting literature.
3. If there are still samples available, it is recommended to add metabolomics data to better explore the interactions between the microbiome and the host.

Re: mSystems00380-25 (Temporal Dynamics of the Vaginal Microbiome and Host Immune Markers Before, During, and After Metronidazole Treatment for Bacterial Vaginosis)

Dear Editor,

We would like to thank you and both reviewers for the constructive comments and important insights regarding the manuscript. Below, please find the reviewer's comments in black *italic* and our responses in red. Line numbers refer to those in the marked-up version of the manuscript (BV_Dynamics_Manuscript_05072025_MARKED.docx). In this submission, we believe we have addressed all comments and concerns through using more descriptive y-axis labels and legends for the appropriate figures, providing additional references where suggested, and reiterating the small sample size in the discussion. If you or either reviewer does not feel we have adequately addressed any comment, we would welcome clarifications or additional discussion to improve the manuscript further.

Reviewers' Comments to Authors:

Reviewer: 1

General comments:

Williams et al provide new results of paired vaginal microbiome (metagenomes) and cytokines in a human cohort of individuals presenting with symptomatic bacterial vaginosis that received oral metronidazole treatment. Metronidazole treatment has been the standard of care for symptomatic bacterial vaginosis for over 50 years, however, recurrent BV is common suggesting treatment is insufficient to provide long term resolution. While the exact etiology is not well-understood, BV is thought to arise from a complex dysbiotic interaction between the vaginal microbiota and vaginal immune response that varies across individuals. Although a small cohort (n =7), this study provides novel characterization of the vaginal microbiome, panbacterial abundance, and vaginal cytokine profile longitudinally prior to, during, and after metronidazole treatment. The authors find a total decrease in bacterial abundance over metronidazole treatment, predominantly driven by reduced BV-associated taxa. Concordantly, they observe an increase in D-lactate producing species during treatment, with no significant microbial features at the end of treatment. In line with prior studies, they find an inflammatory vaginal signature upon BV onset. Interestingly, they found that vaginal sEcad and IP-10 levels were primarily explained by microbial features, and their results confirm the utility of sEcad and IP-10 as markers of molecular BV. Overall, the manuscript was well-written and the experimental design and analyses appropriate. Results and discussion incorporate prior work and study limitations. This work contributes meaningfully to the field of bacterial vaginosis and reveals unrecognized host/microbial kinetics during metronidazole treatment. My comments in general are minor and meant to improve the clarity of the work.

Specific comments:

1. *Metagenomic data was obtained, but relationships beyond taxonomic abundances were not assessed. Were there any functional/genetic associations observed? Can the authors comment on this? How did the genetic signatures of the same CSTs within an individual change over time?*

*In this study, we utilized metagenomic sequencing specifically to achieve higher taxonomic resolution than is possible with 16S rRNA gene sequencing, which cannot distinguish between many closely related species or identify *Gardnerella* genomospecies. This method also provided more accurate species counts to enable us to calculate absolute abundances, providing a more quantitative*

and detailed view of microbiome composition over time. While we did not perform functional or genetic pathway analyses in this manuscript due to the small sample size, we agree that such analyses could offer valuable insights into strain-level variation and CST-specific functional shifts. We believe the current manuscript presents a substantial amount of taxonomic and immunologic data across multiple clinically relevant timepoints, and thus provides a strong foundation for future work exploring functional dynamics.

2. *Figure 1. Consider highlighting the BV Dx a bit more - perhaps an arrow just above the points? While I appreciate the amount of information present in this Figure, which is generally readily interpreted - it took a bit of time to scan for the circle shapes representing BV onset.*
We have changed the shape of BV DX days to a star to improve clarity. We agree that this adjustment makes the figure more immediately interpretable and thank the reviewer for helping us enhance its readability.
3. *Figure 2. Why were there no day 7 points captured? There appears to be one immune point in Figure 4. Please comment. Also, please indicate in the Figure itself (y-axis) that data is log transformed.*
Unfortunately, no day 7 samples were available for analysis. This was due to a typo when plotting the data and has been corrected. We have checked all of the plotting code to ensure no other typos exist for any other plot or analysis. Additionally, we have updated the y-axis labels in Figure 2 to clearly indicate that the data are log-transformed.
4. *Figure 4. The figure legend indicates that the data has been log transformed but this is not indicated in the Figure itself - please clarify.*
. The y-axis label in Figure 4 has been revised to explicitly indicate that the data are log-transformed, ensuring consistency with the figure legend and improving clarity for readers.
5. *Figure 5. Were correlations between cytokines and microbial features statistically significant? This could be indicated in Fig. 5 and/or Supp. Table 6.*
Yes, the correlations shown in Figure 5 were filtered for statistical significance using a p-value threshold of < 0.05 . In response to the reviewer's suggestion, we have updated the figure legend for Figure 5 to clearly indicate this criterion.

Reviewer:

2

General comments:

This study analyzed the dynamics of vaginal microbiota and immune markers in patients with bacterial vaginosis (BV) before, during, and after oral metronidazole (MET) treatment. They revealed that the early treatment is a potential key window for probiotic intervention. In addition, IP-10 and sEcad have been identified as immune markers closely related to the dynamics of BV microbiota, providing new targets for molecular diagnosis and therapeutic monitoring. The results also confirm that metronidazole alone is difficult to restore the protective advantage of lactobacilli, thus emphasizing the necessity of combination therapy (such as probiotics and live biologics).

Major comments:

1. *The final sample size used is relatively small, authors should mention this point in the discussion.*

We agree with the reviewer on this important point. We have added a sentence to the final paragraph of the discussion explicitly acknowledging the relatively small sample size as a limitation, while highlighting the value of the longitudinal design in capturing dynamic changes over the course of treatment. This addition emphasizes the need for future studies with larger cohorts to build upon these findings.

Line 337-339: While the relatively small sample size is a limitation, the study design—featuring longitudinal sampling at key clinical timepoints—enabled a detailed investigation of dynamic microbial and immune shifts in response to treatment.

2. *It is mentioned in the article that the rebound of BV related bacteria after treatment may be related to biofilm, but there is a lack of direct evidence (such as biofilm imaging or gene expression data). It is recommended to supplement relevant discussions or cite supporting literature.. Unfortunately, it was not technically feasible to perform biofilm imaging with the available swab samples. Additionally, generating gene expression data was beyond the scope of this study due to budgetary constraints. However, we believe that the current findings remain robust without these data, as our conclusions are supported by well-established literature. As the reviewer notes, we have already cited several key studies that provide mechanistic insights into the role of biofilms and specific taxa in BV recurrence (see line 254, references 51–53). In light of this comment, we have added additional text and references to contextualize these supporting studies.*

Line 252-257: Furthermore, *Gardnerella vaginalis* is a well-established biofilm-forming species, and studies have shown that its biofilms can persist even after treatment with metronidazole⁵⁰. Evidence suggests that several taxa are capable of integrating into these polymicrobial biofilms⁵¹⁻⁵⁴, where antibiotic penetration is limited, thereby allowing them to persist during treatment and rapidly proliferate post-treatment. Further research is needed to determine whether these taxa contribute to biofilm formation in the vaginal microenvironment.

3. *If there are still samples available, it is recommended to add metabolomics data to better explore the interactions between the microbiome and the host.*

While we agree that metabolomics could offer valuable insights was beyond the scope of this study due to budgetary constraints. To circumvent these limitations and focus on the host side of the interaction, we opted to measure immune markers, which provide a direct and interpretable readout of host responses to microbiome shifts.

Re: mSystems00380-25R1 (Temporal Dynamics of the Vaginal Microbiome and Host Immune Markers Before, During, and After Metronidazole Treatment for Bacterial Vaginosis)

Dear Dr. Johanna B Holm:

Your manuscript has been accepted, and I am forwarding it to the ASM production staff for publication. Before we can proceed, please provide a "Data Availability" paragraph at the end of the Materials and Methods section of full-length articles (or at the end of the text in shorter article types): data description, name of the repository, and DOIs or accession numbers. This policy also applies to any metadata, methods, or algorithms that would be required to replicate the study.

Your paper will first be checked to make sure all elements meet the technical requirements. ASM staff will contact you if anything needs to be revised before copyediting and production can begin. Otherwise, you will be notified when your proofs are ready to be viewed.

Sincerely,
Ying Zhang
Editor
mSystems

Reviewer #1 (Comments for the Author):

The authors have addressed my concerns and the manuscript substantially contributes to the field.

Reviewer #2 (Comments for the Author):

They improved the manuscript and answered my questions.